# Impact of Co-occurrence on Factual Knowledge of Large Language Models

**Cheongwoong Kang**
KAIST
cw.kang@kaist.ac.kr

**Jaesik Choi**
KAIST
jaesik.choi@kaist.ac.kr

## Abstract

Large language models (LLMs) often make factually incorrect responses despite their success in various applications. In this paper, we hypothesize that relying heavily on simple co-occurrence statistics of the pre-training corpora is one of the main factors that cause factual errors. Our results reveal that LLMs are vulnerable to the co-occurrence bias, defined as preferring frequently co-occurred words over the correct answer. Consequently, LLMs struggle to recall facts whose subject and object rarely co-occur in the pre-training dataset although they are seen during finetuning. We show that co-occurrence bias remains despite scaling up model sizes or finetuning. Therefore, we suggest finetuning on a debiased dataset to mitigate the bias by filtering out biased samples whose subject-object co-occurrence count is high. Although debiased finetuning allows LLMs to memorize rare facts in the training set, it is not effective in recalling rare facts unseen during finetuning. Further research in mitigation will help build reliable language models by preventing potential errors. The code is available at https://github.com/CheongWoong/impact_of_cooccurrence.

## 1 Introduction

Natural language processing has seen significant progress in recent years with the advent of large language models (LLMs) (Devlin et al., 2019; Brown et al., 2020; Raffel et al., 2020). Factual knowledge probing benchmarks like LAMA have demonstrated that LLMs have a high capacity to recall factual knowledge (Petroni et al., 2019; Jiang et al., 2020; Roberts et al., 2020; Shin et al., 2020; Zhong et al., 2021). However, factual knowledge stored in LLMs may not always be correct (Elazar et al., 2021; Cao et al., 2021). Understanding the reasons behind such inaccuracies is critical for developing more accurate and reliable language models. Recent studies point out that LLMs often learn short-

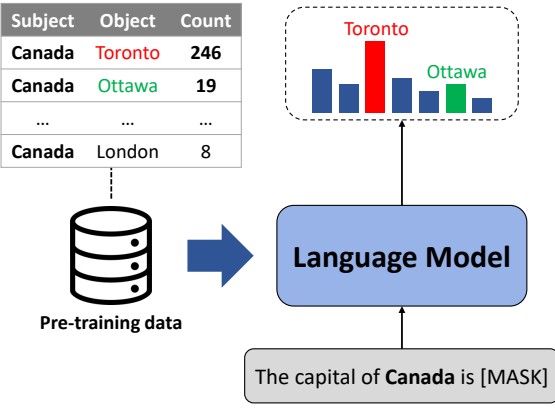

Figure 1: This figure shows an overall framework of our correlation analysis between co-occurrence counts and factual knowledge of LLMs. We assume that if the target model heavily relies on subject-object co-occurrence, it is more likely to recall the most co-occurring word without accurate semantic understanding. For instance, in this hypothetical example, the model fails to answer the question about the capital of Canada by generating the most frequently co-occurring word 'Toronto', while the correct answer is 'Ottawa'. This indicates that relying heavily on co-occurrence statistics may have potential errors.

cuts relying on spurious features rather than understanding language (Wallace et al., 2019; McCoy et al., 2019; Poerner et al., 2020; Ettinger, 2020; Kassner and Schütze, 2020; Cao et al., 2021; Elazar et al., 2021; Bender et al., 2021). We suspect that relying on co-occurrence statistics of the pre-training corpora is one of the main factors that cause such behaviors (Razeghi et al., 2022; Li et al., 2022; Elazar et al., 2022; Kandpal et al., 2023; Kazemi et al., 2023).

In this work, we investigate the effects of co-occurrence statistics of the pre-training data on factual knowledge in LLMs. First, we adopt the LAMA dataset (Petroni et al., 2019) to probe factual knowledge, represented as a subject-relation-object triple. Then, we analyze the correlation between co-occurrence statistics and performance on

factual knowledge probing. Specifically, we count co-occurrences of word pairs in the pre-training corpora. We focus on subject-object co-occurrence, motivated by the concept of distant supervision, which shows that a sentence often contains the triple if it contains a subject and an object of a triple (Mintz et al., 2009). Figure 1 illustrates an overall framework of our correlation analysis between co-occurrence counts and factual knowledge of LLMs. We hypothesize that the target model would generate the most frequently co-occurring word if it heavily relies on co-occurrence. In this simulated example, given the fact 'Canada'-'capital'-'Ottawa', the target model generates the most frequently co-occurring word 'Toronto', which is not the correct answer.

We test our hypothesis with GPT-Neo (Black et al., 2021) and GPT-J (Wang and Komatsuzaki, 2021), which are open-source versions of GPT-3 (Brown et al., 2020). We compute co-occurrence statistics of the Pile dataset (Gao et al., 2020), on which the target models are pre-trained. We show that the factual probing accuracy of LLMs highly correlates with subject-object co-occurrence, leading to failures in recalling rare facts. Although scaling up model sizes or finetuning boosts the overall performance on factual knowledge probing, they do not resolve co-occurrence bias, in which frequently co-occurred words are preferred over the correct answer. Besides, we find that a significant portion of facts in the LAMA dataset can be recalled by simply generating the object with the highest co-occurrence count. Although co-occurrence is necessary to recall factual knowledge, it is not sufficient. Therefore, relying heavily on co-occurrence is inappropriate for understanding the accurate meaning behind words.

Relying heavily on the co-occurrence statistics may lead to hallucinations (Fish, 2009; Maynez et al., 2020; Ji et al., 2023) if the co-occurrence statistics reflect factually incorrect information. Therefore, we suggest finetuning LLMs on the debiased LAMA, constructed by filtering out biased samples whose subject-object co-occurrence count is high. Although finetuning on the debiased dataset allows LLMs to learn rare facts that appear in the training set, it is not generalizable to test cases.

In summary, we show that factual knowledge probing accuracy correlates with subject-object co-occurrence. In addition, we present novel ev- idence and insights by providing a more detailed picture. Specifically, we demonstrate that LLMs prefer frequently co-occurring words, which often override the correct answer, especially when the correct answer rarely co-occurs with the subject. While existing studies only show that the performance of LLMs correlates with co-occurrence, our results provide evidence and reasons for that. We hope our results spur future work on mitigating co-occurrence bias to build accurate and reliable language models.

## 2 Related Work

### 2.1 Prompt Tuning and Finetuning

There have been recent attempts to tune input prompts to improve the performance of LLMs further (Liu et al., 2023b; Lester et al., 2021; Li and Liang, 2021; Qin and Eisner, 2021; Liu et al., 2022, 2023a). However, directly optimizing prompts is not trivial since changes in the input space may cause non-monotonic performance changes (Hu et al., 2022). Especially, Fichtel et al. (2021) demonstrate that finetuned LMs outperform prompt-tuned LMs on factual knowledge probing tasks. Although LLMs, such as GPT-3 and T0, were primitively designed to perform well on various tasks without finetuning (Brown et al., 2020; Sanh et al., 2022), recent work shows that finetuning improves the linguistic capabilities of LLMs substantially (Ouyang et al., 2022; Wei et al., 2022). Therefore, we consider finetuned LMs for analysis.

### 2.2 Term Frequency and Model Behaviors

There have been several approaches to understanding the effects of training data on model behaviors. Specifically, recent studies observe a correlation between pre-training term frequency and model behaviors (Kassner et al., 2020; Wei et al., 2021; Li et al., 2022; Razeghi et al., 2022; Kandpal et al., 2023; Elazar et al., 2022). Our work offers unique contributions by providing additional evidence and insights with in-depth analysis. Specifically, we verify that (1) LLMs learn co-occurrence bias from the pre-training data, preferring frequently co-occurred words over the correct answer, which is especially problematic when recalling rare facts, and (2) co-occurrence bias is not overcome either by scaling up model sizes or finetuning.

## 2.3 Spurious Features

A spurious correlation refers to a relationship in which variables are correlated but does not imply causation due to a coincidence or a confounding factor (Simon, 1954). LMs often learn shortcuts relying on spurious features, such as word overlap, type matching, misprimes, and surface form, which mostly come from dataset bias (Gururangan et al., 2018; McCoy et al., 2019; Wallace et al., 2019; Kassner and Schütze, 2020; Poerner et al., 2020; Wang et al., 2022). For example, if a heuristic (e.g. word overlap, surface form) frequently co-occurs with specific labels, the models may learn the shortcut relying on the heuristic to make decisions. Although spurious features may be helpful in generating plausible responses, it is not appropriate for accurate semantic understanding. Our work suggests that co-occurrence statistics of the pre-training data may work as spurious features, causing hallucinations (Fish, 2009; Maynez et al., 2020; Ji et al., 2023) or biased responses (Bolukbasi et al., 2016; Caliskan et al., 2017; Bommasani et al., 2021).

## 2.4 Memorization

LLMs have been shown to memorize information in training data and generate it verbatim at test time (Emami et al., 2020; Feldman and Zhang, 2020; McCoy et al., 2023; Zhang et al., 2021; Lee et al., 2022; Akyurek et al., 2022; Magar and Schwartz, 2022; Carlini et al., 2023). Memorization implies that LLMs recall memorized information rather than generalizing to new inputs based on learned knowledge. Although recent studies indicate that memorization poses privacy risks (Song and Shmatikov, 2019; Carlini et al., 2019, 2021; Kandpal et al., 2022), it is necessary for near-optimal generalization when learning from a long-tail distribution (Feldman, 2020). Our work also suggests that memorization is essential for accurately recalling facts, since factual knowledge may not be inferred based on prior knowledge of other facts. However, we demonstrate that LLMs often struggle to memorize facts, as the correct answer is overridden by co-occurrence statistics.

## 3 Factual Knowledge Probing

This section describes the overall framework to test factual knowledge of LLMs. With this framework, we aim to investigate the effects of model sizes, finetuning, and co-occurrence statistics.

## 3.1 The LAMA Probe

We adopt the LAMA-TREx dataset (Elsahar et al., 2018; Petroni et al., 2019), which consists of 41 relations, to probe factual knowledge of LLMs. Facts are represented as subject-relation-object triples. Each fact is converted to a natural language form by utilizing a pre-defined set of templates for relations, provided in the original LAMA dataset. For example, a fact 'Canada'-'capital'-'Ottawa' is converted to the sentence "The capital of Canada is Ottawa". Then, each fact is converted to a Cloze statement by masking an object (e.g. "The capital of Canada is [MASK]"), to query the target LM for the masked word. To query unidirectional LMs, we use a sentence truncated right before the mask token (e.g. "The capital of Canada is") in the zero-shot setting while utilizing a full sentence in the finetuned setting. The details of finetuning are included in Appendix A. We assume that the target model knows a fact if it correctly predicts the masked word.

We preprocess the original LAMA-TREx dataset for our experiments. First, we filter out samples whose answer is not in the intersection of the vocabularies of target models. Since the dataset was originally designed for zero-shot knowledge probing, we split the dataset into training and test sets with a ratio of 70:30 to study the effects of finetuning. The data descriptions and statistics, including templates and the number of samples for each relation, are shown in Table 4 in Appendix B.

## 3.2 Metrics

Following the knowledge base completion literature (Bordes et al., 2011, 2013), we consider two rank-based metrics, hits@1 and MRR (mean reciprocal rank), to evaluate the performance on factual knowledge probing. The models that rank ground-truth objects higher are considered more knowledgeable. Hits@1 is 1 if the correct answer is ranked in the top 1 prediction, otherwise 0. MRR is the average reciprocal rank of the correct answer in the prediction. When computing hits@1, we remove other valid objects for a subject-relation pair other than the one we test, following the original setup of LAMA.

## 3.3 Restricted Candidate Sets

Since LLMs are not trained to act as knowledge bases, we use restricted output candidate sets following the recent work (Xiong et al., 2020;

Ravichander et al., 2020; Kassner et al., 2021; Elazar et al., 2021). Specifically, we use three different settings to restrict output vocabularies to study whether LLMs are capable of recognizing appropriate object candidates or not: (1) *remove stopwords*, (2) *gold objects* and (3) *gold objects (relation-wise)*. The *remove stopwords* removes stopwords in the stopword list of NLTK (Bird et al., 2009) from the output candidates. The *gold objects* restricts the output vocabulary to the set of gold objects in the whole dataset. Similarly, the *gold objects (relation-wise)* restricts the output candidates to the set of gold objects for each relation in the dataset.

# 4 Factual Knowledge Probing with Co-occurrence Statistics

This section describes our framework to analyze the impact of pre-training co-occurrence statistics on factual knowledge of LLMs. We first test how much factual knowledge can be recalled with term frequency statistics, including co-occurrence. Then, we analyze the correlation between co-occurrence statistics and factual predictions.

## 4.1 Co-occurrence Statistics

We consider the co-occurrence statistics of the pre-training dataset of target models. Since it is intractable to count co-occurrences of every n-gram pair, we only count co-occurrences between pairs in a minimal sufficient set. This set is initialized as a set of subject entities in the LAMA-TREx dataset and words in the target model's vocabulary, which are object candidates. For text normalization, we tokenize words based on Penn Treebank (Marcus et al., 1993) and remove stopwords in the resulting tokens. Then, we filter out entities those are composed of more than three tokens. Due to computational burdens from the large amount of documents, we count whether an entity pair appears in the same document or not, instead of using a sliding window approach.

## 4.2 Term Frequency Baselines

We test how much factual knowledge can be recalled with simple term frequency statistics by measuring the performance of three different term frequency baselines: (1) *marginal probability*, (2) *joint probability* and (3) *PMI*. For a subject and relation pair, the *marginal probability* baseline ranks object candidates based on how frequently they ap-

pear in the pre-training dataset. The *joint probability* ranks object candidates based on how frequently they appear with the subject in the pre-training dataset. Following the definition of PMI (pointwise mutual information) (Church and Hanks, 1990), the *PMI* baseline normalizes $P_{pretrain}(obj|subj)$, the conditional probability of objects given a subject, by $P_{pretrain}(obj)$, the marginal probability of objects. We measure hits@1 and MRR of the baselines and compare them with the target LLMs.

## 4.3 Correlation Metrics

To analyze the correlation between factual knowledge of LLMs and co-occurrence statistics, we plot hits@1 of the target LLMs against subject-object co-occurrence counts. Here, we consider two types of measures for co-occurrence: (1) the reciprocal rank of subject-object co-occurrence counts and (2) the conditional probability of the gold object given a subject. The former is a relative measure since it considers a reciprocal rank of the gold object among output candidates, while the latter is an absolute measure as it uses conditional probability regardless of other output candidates. Here, we use the co-occurrence statistics of the pre-training corpora to compute co-occurrence counts and conditional probabilities.

# 5 Experiments

## 5.1 Target Models

We test open-source versions of GPT-3 (Brown et al., 2020) with four different model sizes: GPT-Neo 125M, GPT-Neo 1.3B, GPT-Neo 2.7B, and GPT-J 6B (Black et al., 2021; Wang and Komatsuzaki, 2021), which are publicly available on Huggingface's transformers (Wolf et al., 2020). These models are pre-trained on the Pile (Gao et al., 2020), which is a publicly available dataset that consists of 800GB of high-quality texts from 22 different sources.

## 5.2 Results

### 5.2.1 Factual Knowledge Probing

The results of micro-average hits@1 on the test set are reported in Figure 2. Figure 2a shows the results in the zero-shot setting. We observe that hits@1 is higher as the model is larger and as we restrict the output candidates to a smaller set of gold objects. In other words, scaling up model sizes can improve the performance on factual knowledge

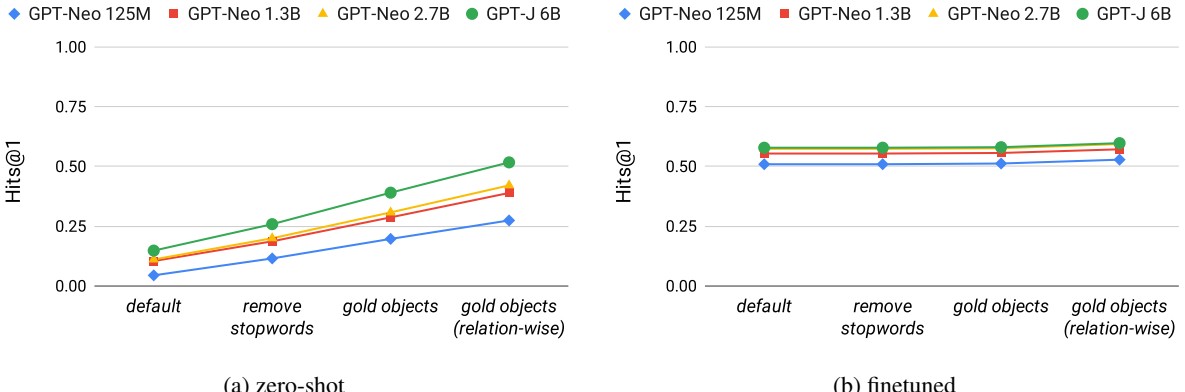

(a) zero-shot        (b) finetuned

Figure 2: **Effects of model sizes and restricted candidate sets:** We plot micro-average hits@1 on the test set. (a) In the zero-shot setting, we observe that hits@1 is higher as the model is larger and as the output vocabulary is restricted to a smaller set. (b) In the finetuned setting, we observe that the effect of model sizes and restricted candidate sets is marginal. **Effects of finetuning:** We observe that finetuning boosts the overall performance.

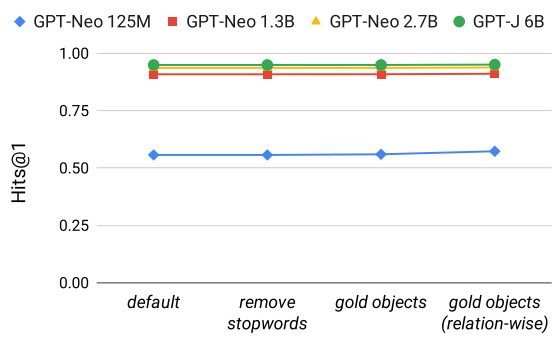

Figure 3: **Memorization capacity of finetuned LLMs:** We show micro-average hits@1 of the finetuned models on the training set. We observe that the models are capable of memorizing most of the seen facts during finetuning except for the smallest model.

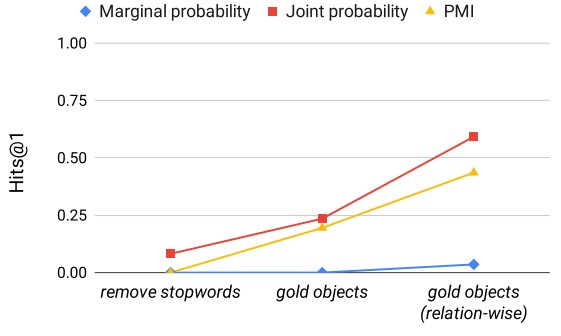

Figure 4: **The results of term frequency baselines:** We report micro-average hits@1 on the test set. We observe that a large portion (about 60%) of the facts can be recalled with the *joint probability* when the output candidates are tightly restricted in the *gold objects (relation-wise)* setting. Note that co-occurrence is useful but not sufficient to recall facts.

probing, and LLMs struggle to recognize appropriate object candidates. Figure 2b presents the results in the finetuned setting. We find that the effect of model sizes is marginal. Different from the zero-shot setting, the effect of restricting the output candidates is also marginal, implying that the models may learn appropriate candidate sets during finetuning. We also observe that finetuning improves factual knowledge probing accuracy substantially. The results of MRR are shown in Figure 10 in Appendix C as they exhibit a similar tendency.

Figure 3 shows the hits@1 results of finetuned models on the training set. We observe that the models except for the smallest one are capable of memorizing most of the seen facts. This implies that memorization is necessary to recall facts since factual knowledge in the test set may not be inferred based on prior knowledge of other facts in the training set.

### 5.2.2 Term Frequency Baselines

Figure 4 shows how much factual knowledge can be recalled with term frequency statistics of the pre-training data. We observe that a large portion (about 60%) of the facts can be recalled with the *joint probability* baseline when the output candidates are tightly restricted in the *gold objects (relation-wise)* setting. The *joint probability* baseline performs as well as GPT-J 6B, the largest model considered in our experiments. The results encourage us to consider the co-occurrence statistics when evaluating language models as it may inflate model performance. Although co-occurrence

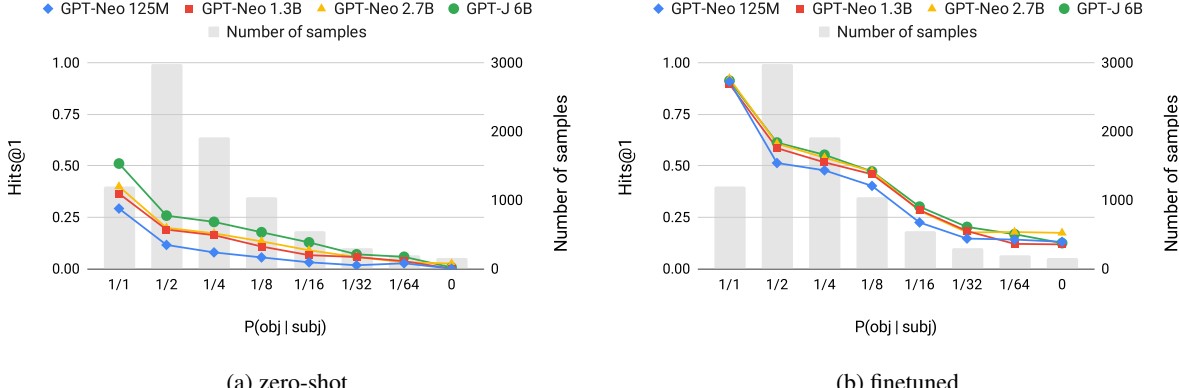

|(a) zero-shot | (b) finetuned|

Figure 5: **The correlation between co-occurrence statistics and factual knowledge probing accuracy**: We plot hits@1 against $P_{pretrain}(obj|subj)$, the conditional probability of the gold object given a subject, on the test set in the *remove stopwords* setting. In both (a) zero-shot and (b) finetuned settings, we observe a strong correlation: hits@1 is lower as the co-occurrence count is lower. As a result, LLMs struggle to recall rare facts. We observe that such correlation remains despite scaling up model sizes or finetuning.

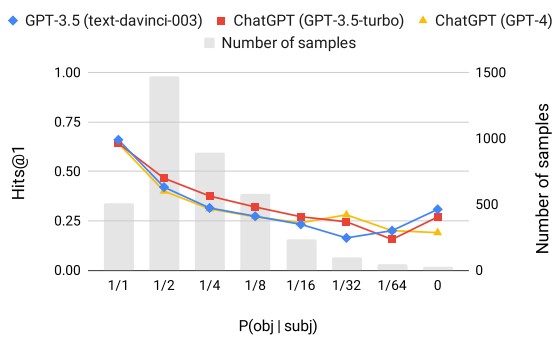

Figure 6: **Correlational analysis of larger models:** We test GPT-3.5 175B and ChatGPT on the subset of test data in the *remove stopwords* setting, verifying that correlation remains despite scaling up model sizes.

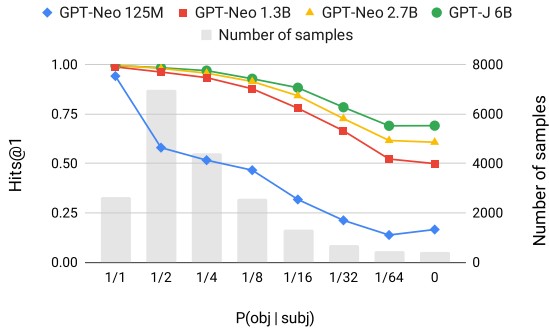

Figure 7: **Correlational analysis of finetuned models on the training set:** We also report the results of fine-tuned models on the training set in the *remove stopwords* setting. Surprisingly, we observe a similar trend to the results on the test set, indicating that LLMs struggle to memorize seen facts if they are rare in the pre-training corpora.

helps recall facts, it may not be appropriate to understand the semantics behind words. The results of MRR, shown in Figure 11 in Appendix C, show a similar tendency.

### 5.2.3 Correlation Analysis

This section reports the results of the correlation between co-occurrence statistics and factual knowledge of LLMs. Note that we exclude facts whose co-occurrence count is unknown (e.g. composed of an entity with more than three tokens), which amounts to less than 6% of the total samples. In this section, we analyze the effects of (1) finetuning and (2) scaling up model sizes.

In Figure 5, we plot hits@1 of the target models against $P_{pretrain}(obj|subj)$, the conditional probability of the gold object given a subject. In both zero-shot and finetuned settings, we observe that hits@1 is lower as the co-occurrence count is lower. Consequently, LLMs suffer from generalizing to recalling rare facts. Comparing Figure 5a and 5b, we observe that finetuning does not resolve co-occurrence bias despite improving the overall performance. In both zero-shot and finetuned settings, we observe that such correlation exists regardless of model sizes. We further test larger models: GPT-3.5 (InstructGPT) 175B (Ouyang et al., 2022) and ChatGPT[12], a GPT optimized to a dialogue system. For ChatGPT, we test two variants with different sizes: (1) GPT-3.5-turbo and (2) GPT-4. Since the vocabulary of ChatGPT is different from the

---
[1] https://openai.com/blog/chatgpt
[2] Note that the pre-training data of GPT-3.5 and ChatGPT are not the same as the Pile, but we use the results as a proxy.

Table 1: The failure cases of GPT-J 6B, preferring words with higher co-occurrence counts over the correct answers. The numbers in parentheses represent the co-occurrence counts.

| Query | Groundtruth | Prediction |
|---|---|---|
| **Tim Mitchell** was born in | *Detroit* (19) | *London* (246) |
| **La Promesse** was created in | *Belgium* (87) | *France* (3420) |
| **Yutaka Abe** died in | *Kyoto* (14) | *Tokyo* (43) |
| **Bell Labs** is owned by | *Nokia* (1744) | *Google* (5167) |
| **Were Ilu** is located in | *Ethiophia* (129) | *Israel* (254) |

Table 2: The quantitative failure analysis of GPT-J 6B, counting how often a word with higher co-occurrence is preferred over the correct answer. We report the ratio of biased cases to the total failure cases in each frequency bin.

| Frequency bin | Ratio |
|---|---|
| 1/1 | 0% |
| 1/2 | 15% |
| 1/4 | 42% |
| 1/8 | 56% |
| 1/16 | 70% |
| 1/32 | 78% |
| 1/64 | 85% |
| 0 | 95% |
| Total | 38% |

Table 3: The failure analysis of GPT-J 6B, comparing the conditional probability of predictions, $P_{pretrain}(\hat{obj}|subj)$, and the conditional probability of the groundtruth objects, $P_{pretrain}(obj|subj)$. We report the mean and standard deviation of conditional probabilities in each frequency bin.

| Frequency bin | $P(\hat{obj}|subj)$ | $P(obj|subj)$ |
|---|---|---|
| 1/1 | 0.42±0.31 | 1.00±0.00 |
| 1/2 | 0.38±0.28 | 0.72±0.14 |
| 1/4 | 0.37±0.27 | 0.37±0.07 |
| 1/8 | 0.31±0.26 | 0.18±0.04 |
| 1/16 | 0.29±0.29 | 0.09±0.02 |
| 1/32 | 0.30±0.31 | 0.05±0.01 |
| 1/64 | 0.26±0.32 | 0.02±0.00 |
| 0 | 0.26±0.30 | 0.01±0.00 |
| Total | 0.35±0.29 | 0.46±0.32 |

and biased responses.

We also quantitatively measure how often the correct answer is overridden by a word with higher co-occurrence counts. Here, we count in each question whether the model's generated answer has higher co-occurrence counts than the correct answer when the model fails to answer correctly. We define a biased case as when the correct answer is overridden by a word with higher co-occurrence counts. Table 2 reports the ratio of GPT-J 6B's biased cases to the total failure cases in each frequency bin. We observe that a word with higher co-occurrence counts overrides the correct answer in a total of 38% of the failure cases. The results of different frequency bins indicate that the co-occurrence bias is more problematic when recalling rare facts. Additionally, Table 3 compares the conditional probability of the generated objects, $P_{pretrain}(\hat{obj}|subj)$, and the conditional probability of the gold objects, $P_{pretrain}(obj|subj)$. The results show that LLMs prefer to generate words that co-occur with the subject frequently enough ($P_{pretrain}(\hat{obj}|subj) \geq 0.26$). In other words, recalling rare facts is especially difficult since words with low co-occurrence counts are hardly generated.

# 6 Mitigation

**Debiasing with Undersampling** Considering facts whose subject-object co-occurrence count is high as biased samples, we suggest finetuning LMs on a debiased dataset, constructed by filtering out biased samples from the training set. Given a dataset $D$ with samples $x_i$ and corresponding bias

open-source target models, we report the results on the subset of test data, which filters out samples whose answer is not in the vocabulary of ChatGPT. The results in Figure 6 verify that scaling up model sizes does not resolve co-occurrence bias while improving the overall performance.

In Figure 7, we investigate the results of seen facts during finetuning. Interestingly, LLMs struggle to learn facts that rarely appear in the pre-training corpora although they are explicitly given during finetuning. The results of hits@1 against the reciprocal rank of subject-object co-occurrence counts are shown in Figure 12, 13 in Appendix C.

## 5.2.4 Failure Analysis

Co-occurrence statistics are necessary but not sufficient to recall facts, as shown in Figure 4. Therefore, a heavy reliance on co-occurrence may be problematic. Table 1 showcases failure cases of GPT-J 6B in the *gold objects (relation-wise)* setting. The examples demonstrate that the model fails to recall facts by selecting words with higher co-occurrence counts over the correct answers. This implies that co-occurrence statistics may often work as spurious features, leading to hallucinations

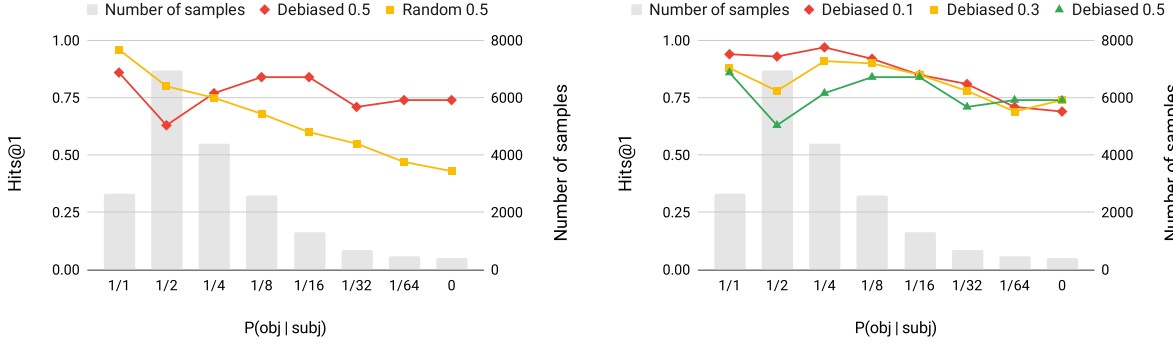

(a) Debiased finetuning vs *random filtering*     (b) Effects of filtering ratios on debiased finetuning

Figure 8: **(a) Effects of debiased finetuning:** We report the micro-average hits@1 of GPT-J 6B on the original training set in the *remove stopwords* setting, comparing debiased finetuning and the *random filtering* baseline. The results show that debiased finetuning helps to learn rare facts but hampers learning frequent facts. **(b) Effects of filtering ratios:** We observe that higher filtering ratios cause performance degradation on frequent facts but marginal improvement on rare facts. Therefore, a filtering ratio needs to be properly tuned to maximize the performance gains on rare facts while keeping the performance on frequent facts.

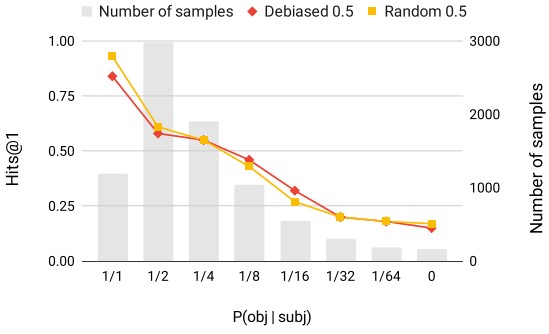

Figure 9: **Effects of debiased finetuning on the test set:** We also analyze the effects of debiased finetuning on the test set. The results show that the effect of debiased finetuning is marginal at test time.

scores $score_{bias}(x_i)$, and a filtering ratio $p$, we discard $p\%$ of the total samples with the highest bias scores. We compute $score_{bias}(x_i)$ as the conditional probability $P_{pretrain}(obj_i|subj_i)$. Since the number of samples is reduced, we consider a *random filtering* baseline that randomly filters out $p\%$ of the total samples for a fair comparison. Note that we apply the filtering algorithms for each relation separately to prevent discarding nearly all samples of a highly biased relation. We test three different filtering ratios: 0.1, 0.3, and 0.5.

The results of GPT-J 6B on the original training set are shown in Figure 8. Figure 8a compares the debiased model with the *random filtering* baseline with a filtering ratio 0.5. The debiased model surpasses the baseline on rare facts with sacrifices on frequent ones. Figure 8b compares the effects

of different filtering ratios on debiased finetuning. We observe that the performance on frequent facts significantly degrades while improvements on rare ones are marginal as more samples are filtered out. Furthermore, we investigate the effects of debiased finetuning on the test set in Figure 9. We observe that the performance of the debiased model and the baseline are similar regardless of co-occurrence counts, implying that the effect of debiased finetuning is not generalizable. Since it is non-trivial to directly fix the cause of co-occurrence bias, designing a more sophisticated debiasing algorithm or using other approaches may be beneficial to complement the proposed debiased finetuning. We leave further investigation in this direction as future work.

## 7  Conclusion

In this work, we investigate the impact of co-occurrence statistics of the pre-training corpora on factual knowledge of LLMs. We reveal the co-occurrence bias of LLMs, in which they prefer words that frequently co-occur with the subject entity over the correct answer. As a result, LLMs struggle to recall facts whose subject and object rarely co-occur in the pre-training corpora although they are seen during finetuning. Although scaling up model sizes or finetuning substantially improves the overall performance, the co-occurrence bias remains. Therefore, we suggest further investigation on mitigating co-occurrence bias to ensure the reliability of language models.

## Limitations

Due to the requirement of large computational resources and the availability of pre-training corpora, we are only able to test a limited set of LLMs. Although we believe that testing other LLMs does not invalidate our claims, it would be good to verify the scalability of the results to strengthen our claims. Another limitation is that our work only focuses on a factual knowledge probing test, which may not be aligned with real-world scenarios. It would be beneficial to investigate how our results generalize to downstream tasks, such as question answering and summarization.

## Ethics Statement

In light of the growing prevalence of LLMs, ethical concerns and challenges have also emerged. For example, LLMs often generate factually incorrect or biased responses. Within this context, our work shows that LLMs strongly correlate with simple co-occurrence statistics of the pre-training corpora, implying that they may generate the most co-occurring word without truly understanding the meaning behind words. We believe that our work has a positive social impact as it suggests a direction toward mitigating potential harms to ensure reliability.

## Acknowledgements

This work was supported by Institute of Information & communications Technology Planning & Evaluation (IITP) grant funded by the Korea government(MSIT) (No.2022-0-00984, Development of Artificial Intelligence Technology for Personalized Plug-and-Play Explanation and Verification of Explanation), (No.2022-0-00184, Development and Study of AI Technologies to Inexpensively Conform to Evolving Policy on Ethics), (No.2019-0-00075, Artificial Intelligence Graduate School Program(KAIST))

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

## A Details of Finetuning

We train models for 3 epochs with a learning rate of 2e-5 and a batch size of 128, padding sequences to a fixed length of 128. Models are trained to predict the masked word only, rather than the whole input prompt. The other hyperparameters are the same as the default hyperparameters of a training script of causal language modeling in Huggingface's transformers (Wolf et al., 2020).

For finetuning uni-directional LMs, we use a manually designed prompt, "### Input:\n {X}\n\n### Response:", where X is replaced with a masked fact (e.g. "The capital of Canada is [MASK] .").

## B Factual Knowledge Probing Dataset

Table 4: Descriptions and statistics of the LAMA-TREx dataset.

| Relation ID | Label | Template | Type | Train | Test |
|---|---|---|---|---|---|
| P17 | country | [X] is located in [Y] . | N-1 | 650 | 262 |
| P19 | place of birth | [X] was born in [Y] . | N-1 | 537 | 243 |
| P20 | place of death | [X] died in [Y] . | N-1 | 582 | 235 |
| P27 | country of citizenship | [X] is [Y] citizen . | N-M | 691 | 267 |
| P30 | continent | [X] is located in [Y] . | N-1 | 657 | 302 |
| P31 | instance of | [X] is a [Y] . | N-M | 608 | 274 |
| P36 | capital | The capital of [X] is [Y] . | 1-1 | 330 | 141 |
| P37 | official language | The official language of [X] is [Y] . | N-1 | 620 | 280 |
| P39 | position held | [X] has the position of [Y] . | N-M | 330 | 155 |
| P47 | shares border with | [X] shares border with [Y] . | N-M | 448 | 203 |
| P101 | field of work | [X] works in the field of [Y] . | N-M | 409 | 164 |
| P103 | native language | The native language of [X] is [Y] . | N-1 | 635 | 284 |
| P106 | occupation | [X] is a [Y] by profession . | N-M | 569 | 252 |
| P108 | employer | [X] works for [Y] . | N-M | 274 | 104 |
| P127 | owned by | [X] is owned by [Y] . | N-1 | 424 | 195 |
| P131 | located in the administrative territorial entity | [X] is located in [Y] . | N-1 | 535 | 240 |
| P136 | genre | [X] plays [Y] music . | N-1 | 616 | 243 |
| P138 | named after | [X] is named after [Y] . | N-1 | 327 | 140 |
| P140 | religion | [X] is affiliated with the [Y] religion . | N-1 | 299 | 135 |
| P159 | headquarters location | The headquarter of [X] is in [Y] . | N-1 | 565 | 236 |
| P176 | manufacturer | [X] is produced by [Y] . | N-1 | 666 | 291 |
| P178 | developer | [X] is developed by [Y] . | N-M | 411 | 177 |
| P190 | twinned administrative body | [X] and [Y] are twin cities . | N-M | 454 | 217 |
| P264 | record label | [X] is represented by music label [Y] . | N-1 | 43 | 10 |
| P276 | location | [X] is located in [Y] . | N-1 | 515 | 251 |
| P279 | subclass of | [X] is a subclass of [Y] . | N-1 | 623 | 280 |
| P361 | part of | [X] is part of [Y] . | N-1 | 533 | 223 |
| P364 | original language of film or TV show | The original language of [X] is [Y] . | N-1 | 531 | 225 |
| P407 | language of work or name | [X] was written in [Y] . | N-1 | 598 | 259 |
| P413 | position played on team / speciality | [X] plays in [Y] position . | N-1 | 675 | 277 |
| P449 | original network | [X] was originally aired on [Y] . | N-1 | 585 | 223 |
| P463 | member of | [X] is a member of [Y] . | N-M | 153 | 50 |
| P495 | country of origin | [X] was created in [Y] . | N-1 | 652 | 253 |
| P527 | has part | [X] consists of [Y] . | N-M | 661 | 295 |
| P530 | diplomatic relation | [X] maintains diplomatic relations with [Y] . | N-M | 667 | 283 |
| P740 | location of formation | [X] was founded in [Y] . | N-1 | 599 | 244 |
| P937 | work location | [X] used to work in [Y] . | N-M | 592 | 261 |
| P1001 | applies to jurisdiction | [X] is a legal term in [Y] . | N-M | 461 | 203 |
| P1303 | instrument | [X] plays [Y] . | N-M | 352 | 161 |
| P1376 | capital of | [X] is the capital of [Y] . | 1-1 | 120 | 59 |
| P1412 | languages spoken, written or signed | [X] used to communicate in [Y] . | N-M | 665 | 259 |
| Total | | | | 20662 | 8856 |

## C   Additional Results

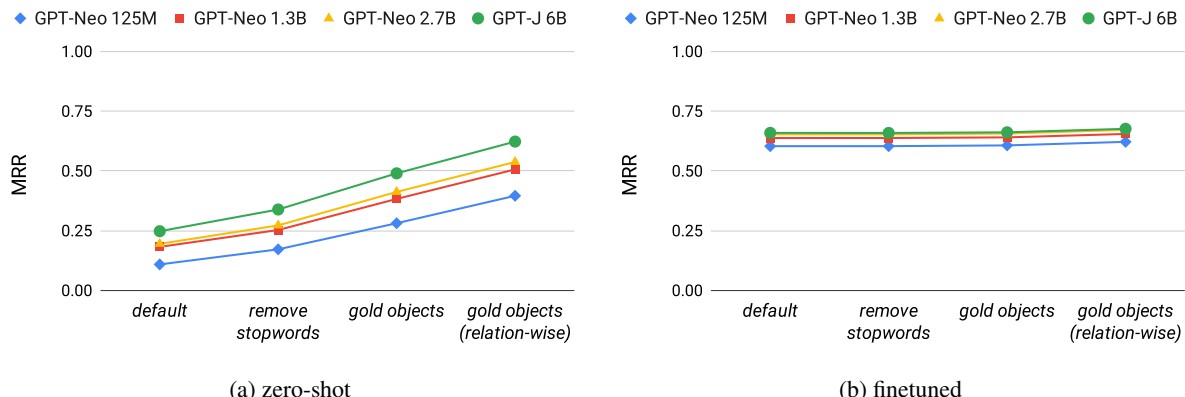

(a) zero-shot

(b) finetuned

Figure 10: **Effects of model sizes and restricted candidates sets:** We show micro-average MRR on the test set. (a) In the zero-shot setting, we observe that MRR is higher as the model is bigger and as the output vocabulary is restricted to a smaller set. (b) In the finetuned setting, we observe that the effect of model sizes and restricted candidate sets is marginal. **Effects of finetuning:** We observe that finetuning boosts MRR on factual knowledge probing.

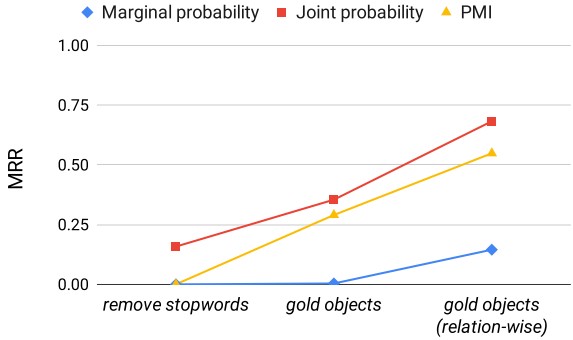

Figure 11: **The results of term frequency baselines:** We plot micro-average MRR on the test set. In the *gold objects (relation-wise)* setting, we observe that the *joint probability* performs as well as GPT-J 6B, which is the largest model considered in our experiments.

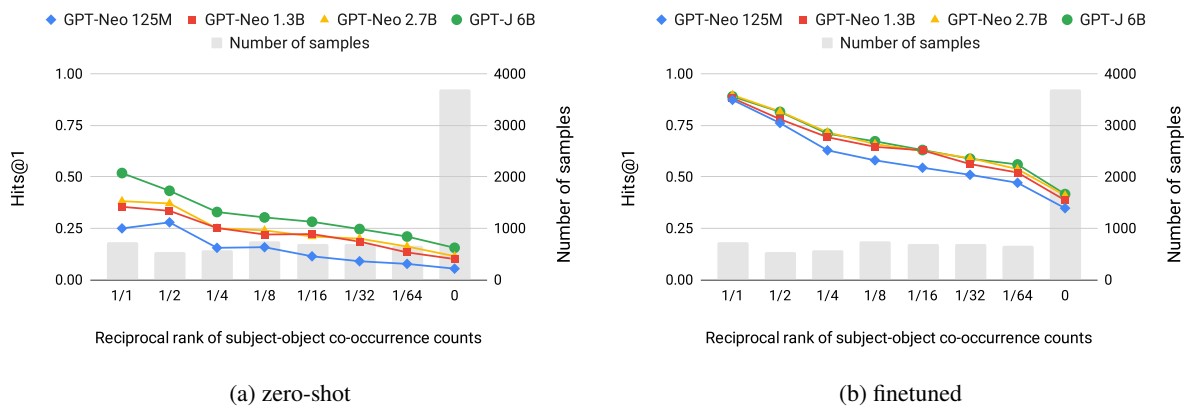

(a) zero-shot

(b) finetuned

Figure 12: **The correlation between co-occurrence statistics and factual knowledge probing accuracy**: We plot hits@1 against the reciprocal rank of subject-object co-occurrence counts on the test set in the *remove stopwords* setting. In both settings (a) and (b), we observe a strong correlation between co-occurrence and hits@1.

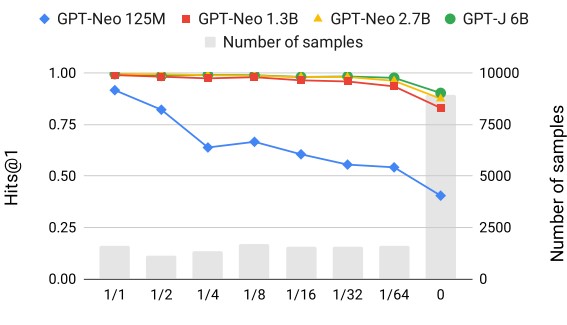

Figure 13: **Correlational analysis of finetuned models on the training set:** We also plot hits@1 of finetuned models against the reciprocal rank of subject-object co-occurrence counts on the training set in the *remove stopwords* setting. Surprisingly, we observe a similar trend to the results on the test set, indicating that LLMs struggle to memorize seen facts if they are rare in the pre-training corpora.