# OpenReview forum: "Impact of Co-occurrence on Factual Knowledge of Large Language Models"
_EMNLP/2023/Conference — EMNLP 2023 Findings_

### Official Review · Reviewer_2dAt · 2023-07-31

**Soundness:** 3

**Excitement:**

3: Ambivalent: It has merits (e.g., it reports state-of-the-art results, the idea is nice), but there are key weaknesses (e.g., it describes incremental work), and it can significantly benefit from another round of revision. However, I won't object to accepting it if my co-reviewers champion it.

**Missing References:**

--

**Paper Topic And Main Contributions:**

The paper investigates the effect of co-occurrence statistics on the ability of large language models to correctly answer simple factual questions (of the subject-relation-object form). The paper specifically checks whether simple co-occurrences between the subject and the object in the pretraining data, can lead the models to incorrectly answer factual questions where the co-occurrence diverges from the correct answer. As it is difficult to check this causal relation by direct manipulation (given the enormous costs of pretraining large language models), a correlation study was conducted. Results show correlation between the co-occurrence statistics of a triplet and the ability of the model to answer correctly questions on it.
The paper further explores mitigation strategies to combat this bias. Two approaches are explored: debiased finetuning and knowledge editing. The former approach presents limited gains. Knowledge editing does show promise, but may be restricted as it requires weeding out the incorrect facts one by one (if I understand correctly). It may also cause unintended changes to unrelated facts.

**Questions For The Authors:**

-- I would appreciate your response to my first point above.

-- Was there any attempt to look at stronger, proprietary models? I'm asking because they are much stronger and may exhibit diff behavior.

-- The point about memorization is unclear to me. Why does co-occurrence imply memorization? on the contrary, if the model "saw" the correct answer and was overridden by co-occurrence this is the opposite of memorization.

-- Why is knowledge editing a mitigation strategy. Obviously wrong knowledge should be corrected, but this is independent of the cause of the mistake, which is the subject of this paper, isn't it?

**Reasons To Accept:**

-- The paper addresses an important and timely topic, namely the ability of LLMs to act as knowledge bases.

-- The related work section is very elaborate and provides insight into the field at hand.

-- The arguments of the paper are well-presented, and the writing is generally clear.

**Reasons To Reject:**

-- I am somewhat confused as to the exact claim the paper is making. While the results clearly show a correlation between the co-occurrence statistics of a triplet and the performance of the model on it, it is not clear to me whether this in fact proves that there is a bias where such simple surface statistics push the language model astray from making the right prediction. What it does show is that questions where the degree of co-occurrence is smaller are more difficult for the model. The introduction reads “in which frequently co-occurred words are preferred over the correct answer.” I could not see how the experiments directly make this point. Simple correlation seems to me insufficient in this case, since making mistakes with little co-occurrence doesn’t mean that there is a different option with higher co-occurrence.

In order to show that the behavior is biased, I would expect the paper to shows that the surface statistics interfere in some sense with the prediction of the model, in a way that would make it predict such answer even when it is not true. For example, I would have expected the paper to examine questions which we would expect (based on their prevalence in the training data) the model to answer correctly, and show that in these cases it tends to make more errors where there is a strong collocation and that the mistakes is towards the collocating words. If the paper indeed makes this kind of more subtle claim and I have missed it, I would welcome a response from the authors on this matter. Thank you.
Following rebuttal: the results you have posted are helpful and address this comment. Please include them in the next version.

-- The results of the attempts to mitigate the bias are not very strong. I should say that I do not see it in itself as grounds for rejection.

-- Some important presentational details are not sufficiently clear (see below).

**Reproducibility:**

5: Could easily reproduce the results.

**Reviewer Confidence:**

4: Quite sure. I tried to check the important points carefully. It's unlikely, though conceivable, that I missed something that should affect my ratings.

**Typos Grammar Style And Presentation Improvements:**

Local comments:
-- What was claimed exactly in previous work with respect to the discussed bias and how does that differ from the work here?

-- l. 136: you write "Our work is the 136 first to investigate the effects of finetuning on the 137 correlation between term frequency statistics and 138 factual knowledge of LLMs." This was not sufficiently clear to me:  I thought your main claim is about the relation between these stats in pretraining and in model behavior. This should be made clearer. Otherwise, a well written previous work section.

-- Section 4.2: Why not use the more standard names then like marginal probability, joint probability and PMI?

-- Figures 2b, 3: The graphs do not show much in my opinion. Could they be explained in a sentence in the text? what does the figure here contribute?

-- Figure 5 (and in other places in the paper): can you also compute correlation w/o binning? what does that turn out to be?

-- Section 6.1: a more formal definition of the filtering method should be given.


Grammar:

-- l. 442: performances s.b. performance

-- l. 501: changes s.b. change

---

> ### Author Rebuttal · Authors · 2023-08-28
>
> Thank you for the constructive reviews.
>
> > What it does show is that questions where the degree of co-occurrence is smaller are more difficult for the model.
>
> We count in each question whether the model's generated answer has a higher co-occurrence count than the ground-truth when the model fails to answer correctly. We found that "a word with higher co-occurrence is preferred over the correct answer" in 38% of the failure cases by GPT-J 6B. We provide the results of different frequency bins below, showing that the co-occurrence bias is more problematic when recalling rare facts. We observe a similar trend for the other models.
> |frequency bin|1/1|1/2|1/4|1/8|1/16|1/32|1/64|0|total|
> |---|---|---|---|---|---|---|---|---|---|
> ||0%|15%|42%|56%|70%|78%|85%|95%|38%|
>
> Below, we also report the results of conditional probability (mean$\pm$std) of predictions (first row) and ground-truths (second row). Since "LLMs prefer words that frequently co-occur with the subject," recalling rare facts is especially difficult.
> |frequency bin|1/1|1/2|1/4|1/8|1/16|1/32|1/64|0|total|
> |---|---|---|---|---|---|---|---|---|---|
> |**$P(\hat{obj}\|subj)$**|0.42$\pm$0.31|0.38$\pm$0.28|0.37$\pm$0.27|0.31$\pm$0.26|0.29$\pm$0.29|0.30$\pm$0.31|0.26$\pm$0.32|0.26$\pm$0.30|0.35$\pm$0.29|
> |**$P(obj\|subj)$**|1.00$\pm$0.00|0.72$\pm$0.14|0.37$\pm$0.07|0.18$\pm$0.04|0.09$\pm$0.02|0.05$\pm$0.01|0.02$\pm$0.00|0.01$\pm$0.00|0.46$\pm$0.32|
>
> > Was there any attempt to look at stronger, proprietary models?
>
> We observed that OpenAI GPT-4 follows a general trend in the correlation between co-occurrence and accuracy, as shown below.
> |frequency bin|1/1|1/2|1/4|1/8|1/16|1/32|1/64|0|
> |---|---|---|---|---|---|---|---|---|
> |**hits@1**|0.64|0.40|0.31|0.27|0.24|0.28|0.20|0.19|
>
> > The point about memorization is unclear to me. Why does co-occurrence imply memorization?
>
> We agree that "relying on co-occurrence implies simple memorization" may be erroneous. We would rather say "LLMs often memorize surface statistics to recall facts, which may not be appropriate for actual semantic understanding.
>
> > Why is knowledge editing a mitigation strategy?
>
> It is non-trivial to directly fix the cause of the mistakes, as suggested in the results of Section 6.1 (finetuning on a balanced dataset). Therefore, we tested knowledge editing as an alternative among various candidates (probability calibration, retrieval) and showed its effectiveness. Since mitigation is not our main focus, we leave further investigation in this direction as future work.
>
> > What was claimed exactly in previous work with respect to the discussed bias and how does that differ from the work here?
>
> "The correlation between word frequency and model behaviors" is previously observed. However, the paper offers unique contributions by providing additional evidence and insights with detailed analysis. We verify that (1) LLMs learn co-occurrence bias from the pre-training data, which is especially problematic when recalling rare facts, and (2) co-occurrence bias is not perfectly overcome by scaling up model sizes or finetuning. The observations question whether LLMs recall facts with actual semantic understanding or not.
>
> > "Our work is the first to investigate the effects of finetuning on the correlation between term frequency statistics and factual knowledge of LLMs."
>
> Since our main claim is not "to investigate the effects of finetuning on the correlation between term frequency statistics and factual knowledge of LLMs," we decided to rewrite this part to highlight the main claim.
>
> > Figures 2b, 3: The graphs do not show much in my opinion. Could they be explained in a sentence in the text? what does the figure here contribute?
>
> The figures show the effects of finetuning on the overall performance. Figure 2b shows that finetuning improves the performance substantially. Figure 3 shows that the models (except for the smallest one) memorize most of the facts in the finetuning dataset. We tried to show that the co-occurrence bias remains (Figures 5b, 7) after finetuning, although it improves the overall performance by a large margin (Figures 2b, 3).
>
> > Figure 5 (and in other places in the paper): can you also compute correlation w/o binning? what does that turn out to be?
>
> We computed the pearson correlation without binning. We observed a positive correlation to some extent (p-value < 0.001), and the magnitude without binning is similar to that with binning.
> ||w/o binning|with binning|randomly shuffled|
> |---|---|---|---|
> |**GPT-J 6B**|0.24|0.24|0.01|
> |**GPT-J 6B (finetuned)**|0.35|0.36|0.00|
>
> > Other comments: Section 4.2: Why not use the more standard names then like marginal probability, joint probability and PMI? Section 6.1: a more formal definition of the filtering method should be given.
>
> We will reflect the suggestions in the revised version.

---

### Official Review · Reviewer_B1hy · 2023-08-05

**Soundness:** 3

**Excitement:**

2: Mediocre: This paper makes marginal contributions (vs non-contemporaneous work), so I would rather not see it in the conference.

**Paper Topic And Main Contributions:**

The paper argues that large language models suffer from the so-called 'co-occurrence bias', i.e. when asked a factual question they tend to assign higher probabilities to words that have higher co-occurrence statistics with the query instead of the correct answer. The authors show that this bias persists in models of different sizes and cannot be effectively removed using fine-tuning, but can be partially remedied with knowledge editing. In order to prove their point, the authors introduce a series of term-frequency baselines, two of which are co-occurrence statistics. They show that these statistics can be used as a shortcut feature (in some settings, selecting the most frequently co-occurrent word results in 60% accuracy on the test set) and explain the actual behaviour of the models to a large extent. The paper explores two mitigation strategies: fine-tuning on a debiased version of the training dataset and rank-one model editing (ROME). While fine-tuning is not particularly effective, ROME shows promising results on frequent relation-like facts but does not really help with rarer ones.

**Questions For The Authors:**

In the discussion of fine-tuning in lines 347--348, it is said that "the models may learn appropriate cadidate sets during finetuning." Does this mean that there is some amount of knowledge leakage in fine-tuning and that the testing set-up is slightly different from the zero-shot setting?

In Figure 6, we see that the accuracy of the models' outputs _improves_ when the conditional probability of the object given the subject goes down from around 1/64 to 0. This goes against the general trend of the positive association between conditional probability and accuracy. Is there any interpretation for this?

**Reasons To Accept:**

In my opinion, the most interesting aspect of the paper are the frequency baselines introduced to analyse the structure of the training dataset and to explain the behaviour of the models. They bring the co-occurrence/frequency bias influencing the behaviour of the models to fore, and the analysis is convincing.

**Reasons To Reject:**

The main point of the paper -- that LLMs suffer from co-occurrency bias -- is not particularly new. Other papers investigating this issue are mentioned in the Related Work section. The authors claim that their work is the fist "to investigate the effects of finetuning on the correlation between term frequency statistics and factual knowledge of LLMs" (ll. 136--139), but there is no discussion of why fine-tuning should help at all, and in the end it does not, which amounts to a weak negative result. The section on mitigating occupies less than 1.5 pages and does not contain any methodological insights.

**Reproducibility:**

3: Could reproduce the results with some difficulty. The settings of parameters are underspecified or subjectively determined; the training/evaluation data are not widely available.

**Reviewer Confidence:**

4: Quite sure. I tried to check the important points carefully. It's unlikely, though conceivable, that I missed something that should affect my ratings.

**Typos Grammar Style And Presentation Improvements:**

Some of the statements in the paper look confusing. In lines 87--89, the authors point out that "relying heavily on co-occurrence is not appropriate for understanding the accurate meaning behind words." This is probably true, but this is all we have when training LLMs, and the paper does not propose a way out. Similarly, in lines 173--174, the authors state that "relying on co-occurrence implies simple memorization." Again, the relationship between co-occurrence and memorization is more complicated, and equating one with the other seems erroneous.

---

> ### Author Rebuttal · Authors · 2023-08-28
>
> Thank you for the valuable reviews.
>
> > The main point of the paper -- that LLMs suffer from co-occurrence bias -- is not particularly new.
>
> We agree that "the correlation between word frequency and model behaviors" is previously observed. However, the paper offers unique contributions by providing additional evidence and insights with detailed analysis. We verify that (1) LLMs learn co-occurrence bias from the pre-training data, which is especially problematic when recalling rare facts, and (2) co-occurrence bias is not perfectly overcome by scaling up model sizes or finetuning. The observations question whether LLMs recall facts with actual semantic understanding or not.
>
> > No discussion of why fine-tuning should help at all, and in the end it does not.
>
> Since our main claim is not "to investigate the effects of finetuning on the correlation between term frequency statistics and factual knowledge of LLMs", we decided to rewrite this part to highlight the main claim. Besides that, we analyze the effects of finetuning since one may think that LLMs learn something beyond surface statistics during finetuning as it significantly improves the overall performance.
>
> > "the models may learn appropriate candidate sets during finetuning."
>
> No facts overlap between the finetuning and test datasets. However, the distribution of possible objects for each relation P(o|r) may be learned during finetuning. Although this can be thought as data leakage to some extent, it is non-trivial to recall the facts solely based on P(o|r).
>
> > In Figure 6, some results against the general trend of the positive association between conditional probability and accuracy.
>
> This may be because (1) the pre-training datasets of these models (OpenAI GPT) are different from those of GPT-Neo/J, and (2) there are only a few samples whose conditional probability is 0.
>
> > The section on mitigating occupies less than 1.5 pages and does not contain any methodological insights. The paper does not propose a way out.
>
> Our main focus is to analyze the relationships between "co-occurrence bias" and model behaviors. Although we only showed that knowledge editing may be effective in mitigating the problem, there may be other possible approaches (probability calibration, retrieval). We leave further investigation in this direction as future work.
>
> > The authors state that "relying on co-occurrence implies simple memorization." Again, the relationship between co-occurrence and memorization is more complicated, and equating one with the other seems erroneous.
>
> We agree that "relying on co-occurrence implies simple memorization" may be erroneous. We would rather say "LLMs often memorize surface statistics to recall facts, which may not be appropriate for actual semantic understanding.

---

### Official Review · Reviewer_M9gY · 2023-08-05

**Soundness:** 4

**Excitement:**

3: Ambivalent: It has merits (e.g., it reports state-of-the-art results, the idea is nice), but there are key weaknesses (e.g., it describes incremental work), and it can significantly benefit from another round of revision. However, I won't object to accepting it if my co-reviewers champion it.

**Missing References:**

[1] Chain-of-Thought Prompting Elicits Reasoning in Large Language Models

[2] Identifying and Mitigating Spurious Correlations for Improving Robustness in NLP Models (NAACL Findings 2022)

**Paper Topic And Main Contributions:**

The paper proposes a framework to probe the shortcuts in LLMs. The framework firsts test the accuracy of LLM on factual knowledge dataset like LAMA. Then the framework computes the correlation of the subject and the object in the pretraining corpus. Results on GPT models (from 125M to 6B) show that, as the co-occurrence of subject and object decreases, the accuracy also decreases. Larger models (GPT-3.5 of 175B) also suffer from the problem. The authors also find that simple baseline based on co-occurrence is sufficient to surpass the performance of GPT-J 6B. Finally, the authors also give solution to mitigate the shortcut problem.

**Questions For The Authors:**

-	The paper focuses only on token-level shortcuts. Can the findings be generalized to other kinds of shortcuts, such as word-overlap (Right for the wrong reasons: Diagnosing syntactic heuris- tics in natural language inference) or length (Annotation artifacts in natural language inference data) shortcuts?
-	The proposed mitigation methods include balancing the data and knowledge editing. However, I am curious about whether demonstrations or Chain-of-Thoughts [1] can also solve the issue.
-	Fig. 3 shows that smaller model cannot memorize the data. Then why it also has the shortcut problem like other larger models who memorize the biases in the pretraining data?

**Reasons To Accept:**

-	The paper contains detailed analysis of shortcut problem regarding token co-occurrence.
-	The paper investigates an important area of verifying the factual knowledge of LLMs.
-	The paper is well-written, and is free of significant presentation issues.
-	Along with the identification of the problem, the paper also proposes to mitigate the problem.


**Reasons To Reject:**

-	One more experiment should be done to verify the claim that "answers with higher co-occurrence are more likely to be generated": The authors should count in each question, whether the model's generated answer has a high count in the pretraining corpus. Currently there is only a table (Table 1) showing similar results, i.e., the wrong answers have a relatively lower count in the pretraining corpus. However, quantitative results over the whole dataset should be given.
-	There are existing work probing the shortcut learning problem of language models. The authors should elaborate more on the work to claim that they are the first to investigate the effects of finetuning on the correlation between term frequency statistics and factual knowledge of LLMs. For example: [2].

**Reproducibility:**

4: Could mostly reproduce the results, but there may be some variation because of sample variance or minor variations in their interpretation of the protocol or method.

**Reviewer Confidence:**

4: Quite sure. I tried to check the important points carefully. It's unlikely, though conceivable, that I missed something that should affect my ratings.

---

> ### Author Rebuttal · Authors · 2023-08-28
>
> Thank you for the constructive reviews.
>
> > One more experiment should be done to verify the claim that "answers with higher co-occurrence are more likely to be generated"
>
> We count in each question whether the model's generated answer has a higher co-occurrence count than the ground-truth when the model fails to answer correctly. We found that "a word with higher co-occurrence is preferred over the correct answer" in 38% of the failure cases by GPT-J 6B. We provide the results of different frequency bins below, showing that the co-occurrence bias is more problematic when recalling rare facts. We observe a similar trend for the other models.
> |frequency bin|1/1|1/2|1/4|1/8|1/16|1/32|1/64|0|total|
> |---|---|---|---|---|---|---|---|---|---|
> ||0%|15%|42%|56%|70%|78%|85%|95%|38%|
>
> Below, we also report the results of conditional probability (mean$\pm$std) of predictions (first row) and ground-truths (second row). Since "LLMs prefer words that frequently co-occur with the subject," recalling rare facts is especially difficult.
> |frequency bin|1/1|1/2|1/4|1/8|1/16|1/32|1/64|0|total|
> |---|---|---|---|---|---|---|---|---|---|
> |**$P(\hat{obj}\|subj)$**|0.42$\pm$0.31|0.38$\pm$0.28|0.37$\pm$0.27|0.31$\pm$0.26|0.29$\pm$0.29|0.30$\pm$0.31|0.26$\pm$0.32|0.26$\pm$0.30|0.35$\pm$0.29|
> |**$P(obj\|subj)$**|1.00$\pm$0.00|0.72$\pm$0.14|0.37$\pm$0.07|0.18$\pm$0.04|0.09$\pm$0.02|0.05$\pm$0.01|0.02$\pm$0.00|0.01$\pm$0.00|0.46$\pm$0.32|
>
> > Elaborate more on the work to claim that they are the first to investigate the effects of finetuning on the correlation between term frequency statistics and factual knowledge of LLMs.
>
> As our main claim is not "to investigate the effects of finetuning on the correlation", we decided to rewrite this part to emphasize the main claim.
> We agree that "the correlation between word frequency and model behaviors" is previously observed in existing work. However, the paper offers unique contributions by providing additional evidence and insights with detailed analysis. We verify that (1) LLMs learn co-occurrence bias from the pre-training data, which is especially problematic when recalling rare facts, and (2) co-occurrence bias is not perfectly overcome by scaling up model sizes or finetuning. The observations question whether LLMs recall facts with actual semantic understanding or not.
>
> > Can the findings be generalized to other kinds of shortcuts, such as word-overlap or length?
>
> Yes. For example, if a heuristic (word overlap or sentence length) frequently co-occurs with specific labels, the model may learn the shortcut, preferring labels that frequently co-occur with the heuristics.
>
> > However, I am curious about whether demonstrations or Chain-of-Thoughts can also solve the issue.
>
> We test prompting with demonstrations in a 4-shot setting, where four demonstrations with the same relation are randomly given for each sample. The results of GPT-J 6B in the remove_stopwords setting show significant improvements in hits@1 (**0.26 => 0.55**). However, the co-occurrence bias remains, as shown in the table below.
> |frequency bin|1/1|1/2|1/4|1/8|1/16|1/32|1/64|0|
> |---|---|---|---|---|---|---|---|---|
> |**hits@1**|0.86|0.62|0.51|0.42|0.26|0.15|0.09|0.07|
>
> > Fig. 3 shows that smaller model cannot memorize the data. Then why it also has the shortcut problem like other larger models who memorize the biases in the pretraining data?
>
> Fig. 3 shows that the smallest model cannot memorize the fine-tuning data, not the pre-training data. However, the shortcut problem in this paper arises when a model memorizes the biases (shortcuts) in the pre-training data.

---

### Meta-Review · Area_Chair_8Mj6 · 2023-09-17

**Recommendation:** 3

**Metareview:**

The paper investigates the co-occurrence bias in large language models, i.e. the tendency to assign higher probabilities to words that have higher co-occurrence statistics instead of the correct answer. The paper shows that this bias persists in models of different sizes and cannot be effectively removed using fine-tuning, but can be partially remedied with knowledge editing. While the methodology for studying and mitigating co-occurrence bias is novel, the insights that the paper provides are limited. Some of the claims the paper makes are not substantiated clearly enough, although during the discussion the authors provided some clarifications and additional results that will help to strengthen the paper.

---

### Decision · Program_Chairs · 2023-10-07

**Decision:**

Accept-Findings

**Comment:**

The paper investigates the co-occurrence bias in large language models, i.e. the tendency to assign higher probabilities to words that have higher co-occurrence statistics instead of the correct answer. The paper shows that this bias persists in models of different sizes and cannot be effectively removed using fine-tuning, but can be partially remedied with knowledge editing. While the methodology for studying and mitigating co-occurrence bias is novel, the insights that the paper provides are limited. Some of the claims the paper makes are not substantiated clearly enough, although during the discussion the authors provided some clarifications and additional results that will help to strengthen the paper.